# Sirt6 Activation Ameliorates Inflammatory Bone Loss in Ligature-Induced Periodontitis in Mice

**DOI:** 10.3390/ijms241310714

**Published:** 2023-06-27

**Authors:** Myung Jin Lee, Hyang Hwa Ryu, Jae Won Hwang, Jung Ryul Kim, Eui-Sic Cho, Jin Kyeong Choi, Young Jae Moon

**Affiliations:** 1Department of Chemistry, Case Western Reserve University, Cleveland, OH 44106, USA; jinalee939@gmail.com; 2Department of Biochemistry and Molecular Biology, Jeonbuk National University Medical School, Jeonju 54896, Republic of Korea; yy-ryu@hanmail.net (H.H.R.); toriake11@naver.com (J.W.H.); 3Department of Orthopaedic Surgery, Jeonbuk National University Medical School and Hospital, Jeonju 54896, Republic of Korea; jrkeem@jbnu.ac.kr; 4Cluster for Craniofacial Development and Regeneration Research, Institute of Oral Biosciences, Jeonbuk National University School of Dentistry, Jeonju 54896, Republic of Korea; oasis@jbnu.ac.kr; 5Department of Immunology, Jeonbuk National University Medical School, Jeonju 54907, Republic of Korea

**Keywords:** alveolar bone loss, inflammation, MDL801, mouse models, osteoclast, periodontitis, Sirt6

## Abstract

Periodontitis is an inflammatory disease caused by microorganisms that induce the destruction of periodontal tissue. Inflamed and damaged tissue produces various inflammatory cytokines, which activate osteoclasts and induce alveolar bone loss and, eventually, tooth loss. Sirt6 expression suppresses inflammation and bone resorption; however, its role in periodontitis remains unclear. We hypothesized that Sirt6 has a protective role in periodontitis. To understand the role of Sirt6 in periodontitis, we compared periodontitis with ligature placement around the maxillary left second molar in 8-week-old control (C57BL/6J) male mice to Sirt6-overexpressing Tg (Sirt6Tg) mice, and we observed the resulting phenotypes using micro-CT. MDL801, a Sirt6 activator, was used as a therapy for periodontitis through oral gavage. Pro-inflammatory cytokines and increased osteoclast numbers were observed in alveolar bone tissue under periodontitis surgery. In the same condition, interestingly, protein levels from Sirt6 were the most downregulated among sirtuins in alveolar bone tissue. Based on micro-CT and CEJ-ABC distance, Sirt6Tg was observed to resist bone loss against ligature-induced periodontitis. Furthermore, the number of osteoclasts was significantly reduced in Sirt6Tg-ligated mice compared with control-ligated mice, although systemic inflammatory cytokines did not change. Consistent with this observation, we confirmed that bone loss was significantly reduced when MDL801, a Sirt6 activator, was included in the ligation mouse model. Our findings demonstrate that Sirt6 activation prevents bone loss against ligature-induced periodontitis. Thus, a Sirt6 activator may provide a new therapeutic approach for periodontitis.

## 1. Introduction

Periodontitis is an inflammatory condition resulting from a host immune response to microbial infection, leading to the destruction of connective tissue and alveolar bone, and, ultimately, tooth loss [1]. Emerging evidence has suggested that severe periodontitis is associated with other chronic inflammatory diseases including diabetes, cardiovascular disease, atherosclerosis, and surgical site infection [2,3,4,5]. Additionally, periodontal bacteria can be transmitted by hematogenous, oropharyngeal, and orodigestive pathways, causing or exacerbating inflammatory pathologies such as rheumatoid arthritis, Alzheimer’s disease, aspiration pneumonia, cardiovascular disease, and inflammatory bowel disease [6]. From this perspective, treatment for periodontitis could reduce the risk of local and systemic comorbidities related to periodontitis.

Inflammatory bone loss is the hallmark of periodontitis. Cumulated alveolar bone loss weakens the support structure of the teeth, causing tooth mobility and loss [7]. Many procedures are described in the literature to reduce bone loss and alveolar shrinking after tooth extraction; however, the prevention of tooth loss is the primary objective [8,9,10]. Pathophysiologically, microbial infection promotes bone resorption in the periodontal region by activating immune cells and stromal cells to induce osteoclast formation [2]. Immune cells secrete pro-inflammatory mediators, including IL-1, IL-6, and TNF-α, to activate osteoclasts and attenuate osteoblast activity [3,4,11]. Furthermore, immune and stromal cells trigger bone resorption by stimulating the receptor activator of nuclear factor kappa-Β ligand (RANKL) signaling, which promotes osteoclast formation [6,12]. Clinically, according to the 2017 World Workshop on the Classification of Periodontal and Peri-Implant Disease and Conditions, radiographic bone loss was determined by the stage and degree of periodontitis and was introduced as an indicator to predict disease progression [13]. Therefore, methods to control bone loss by suppressing the inflammatory response or directly regulating osteoclast activity can be a target for treating periodontitis.

Sir2-like protein 6 (Sirt6) is a nuclear NAD^+^-dependent deacetylase to histones and nuclear proteins. Sirt6 plays roles in DNA repair, genome stability, glucose homeostasis, and suppression of the inflammatory response through epigenetic regulation [12,14,15,16]. Accumulating evidence demonstrates that Sirt6 has a protective effect on inflammatory diseases, including rheumatoid arthritis, Alzheimer’s disease, cardiovascular diseases, and hepatic diseases [17]. Furthermore, our previous studies showed that Sirt6 inhibited bone resorption by regulating osteoclast apoptosis and reduced osteoclastogenesis by regulating RANKL expression in osteoblast and osteocyte [18]. These pieces of evidence suggested that the Sirt6 activation may also alleviate symptoms in periodontitis, but there are no related studies yet. Therefore, we hypothesized that Sirt6 is essential in regulating inflammation and bone loss in periodontitis. We also showed a protective effect of Sirt6 against ligature-induced periodontitis using Sirt6 overexpression in mice and a Sirt6 activator.

## 2. Results

### 2.1. Inflammatory Bone Loss with an Increased Osteoclast Number Is Observed in Ligature-Induced Periodontitis

To establish inflammatory bone loss in periodontitis, we used ligature-induced periodontitis. Twelve-week-old WT mice were used, and the mice were sacrificed three weeks after placement of the ligature on the second molar. The WT ligature group was observed to have alveolar bone loss around the ligated molar and changes in oral epithelium thickness when compared to the sham group (Figure 1A,B). Inflammatory cells infiltrated the site of alveolar bone loss (Figure 1B), and expression of pro-inflammatory cytokines (IL-6 and IL-1β) were significantly increased in the WT ligature group (Figure 2A). Changes in these inflammatory cytokines were the same as previously reported [19]. IL-6 and IL-1β are expressed and secreted primarily by neutrophils, monocytes, and macrophages [20], and activate osteoclasts and promote bone resorption [6]. Thus, we evaluated the infiltration of these cells into the periodontal region using CD11b (neutrophils and monocytes) and F4/80 (macrophage) cell markers. As expected, CD11b-positive cells and F4/80-positive cells were significantly increased in the ligature group compared to the sham in the gingiva and periodontal ligament area (Figure 2B). Based on these reports, we confirmed the changes in osteoclasts around the lesion area. As shown Figure 1C, TRAP-positive osteoclast number and osteoclast surface around the alveolar bone surface were significantly increased in the WT ligature group compared with the sham group. However, no systemic inflammatory response was observed (Appendix A). These results suggest that the ligature-induced periodontitis mouse model is similar to the pathophysiologic mechanism of periodontitis.

### 2.2. Sirt6 Expression Is Downregulated in Ligature-Induced Periodontitis

Since sirtuins play an essential role in the infiltration of neutrophils and macrophages via the inflammatory response [17,21], we investigated whether sirtuins are involved in periodontitis. To identify the role of sirtuins in periodontitis, we evaluated the protein level of sirtuins in alveolar bone from ligatured mice for three weeks and compared it with the sham group. Among the sirtuin family proteins, Sirt1 and Sirt7 were observed in a decreased pattern, and Sirt6 in particular showed a statistically significant difference between the WT sham and WT ligature groups (Figure 2C).

### 2.3. Sirt6 Overexpressing Mice Have Resistant to Bone Loss by Ligature-Induced Periodontitis

To verify the role of Sirt6 in alveolar bone loss induced by periodontitis, we used Sirt6-overexpressing Tg mice (Sirt6Tg) [22]. After ligature surgery, mice were sacrificed at one week and three weeks; then, micro-CT analyses were performed in Sirt6Tg and WT mice. Volumetric measurements from micro-CT revealed that bone volume (BV/TV) decreased in both genotypes at one week, whereas at three weeks, Sirt6Tg had significantly increased compared with controls (Figure 3A,C). Furthermore, after three weeks of ligature, Sirt6Tg mice had increased trabecular numbers compared with controls and reduced trabecular separation (Figure 3C). Three-dimensional reconstruction images showed that the distance from the cemento–enamel junction (CEJ) to the alveolar bone crest (ABC) was significantly reduced in Sirt6Tg mice compared with controls three weeks after ligation (Figure 3D). These results suggested that Sirt6Tg resists bone loss against ligature-induced periodontitis compared with controls.

### 2.4. Sirt6Tg Mice Exhibit a Reduction in Osteoclast Activity and Inflammation against Ligature-Induced Periodontitis

Because Sirt6 plays a role in anti-inflammation and inhibition of osteoclast activity [17,22,23], we observed that reduced bone loss in Sirt6Tg mice against ligature-induced periodontitis results in blocking inflammation and osteoclasts. As shown in Figure 4, TRAP-positive osteoclast numbers and osteoclast surface around the alveolar bone surface significantly decreased in the Sirt6Tg ligature mice compared with controls. Additionally, pro-inflammatory cytokine (IL-6 and IL-1β) expression was dramatically reduced in the Sirt6Tg mice (Figure 4C). However, no significant changes were observed in systemic inflammatory response or the systemic osteoclast activation index (Appendix A).

### 2.5. Pharmacological Sirt6 Activation Suppresses Bone Loss by Ligature-Induced Periodontitis

Next, we tested whether pharmacological Sirt6 activation was effective in ligature-induced periodontitis. We administered the Sirt6-specific activator MDL801 for three weeks after the second molar ligation and analyzed the bone phenotype. Micro-CT analysis showed that bone volume (BV/TV), trabecular thickness (Tb.Th), and trabecular numbers (Tb.N) were markedly increased in the MDL801-treated group compared with vehicle (Figure 5A–C). Furthermore, the distance from CEJ to ABC was significantly reduced in MDL801-treated mice compared with vehicle-treated mice (Figure 5D).

## 3. Discussion

This study showed that Sirt6 regulates inflammatory bone loss in ligature-induced periodontitis by suppressing osteoclast number and pro-inflammatory cytokines. These results were verified in genetically Sirt6-overexpressing mice and pharmacologic Sirt6 activation.

Sirt6 had two roles in protecting against bone loss caused by ligature-induced periodontitis. First, Sirt6 had a role in regulating pro-inflammatory cytokines in a model of ligature-induced periodontitis. In the maxillary tissue of ligature-induced periodontitis, Sirt6 expression was downregulated, and inflammatory cell accumulation accompanied by inflammatory cytokines was observed. In contrast, Sirt6-overexpressing mice showed decreased pro-inflammatory cytokines enhanced by ligation compared with the control group. This result is supported by accumulated evidence associated with Sirt6. Sirt6 protein-deficient tissue exhibited ectopic expression of the NF-κB target gene, which is a crucial modulator of pro-inflammatory genes [14]. This mechanism demonstrated that depletion of Sirt6 exacerbates disease by promoting inflammation in collagen-induced arthritis [24], wound healing [25], and high-fat-induced insulin resistance [26] in myeloid-specific Sirt6 knockout mice. Second, Sirt6 had a role in regulating osteoclast number in a model of ligature-induced periodontitis. Similar to previous reports, which showed bone loss due to enhancement of osteoclast number in ligature-induced periodontitis models [27,28], the osteoclasts around the alveolar bone in this study also significantly increased in the periodontitis model. In this disease model, Sirt6 overexpression suppressed the number of osteoclasts around the maxilla and prevented bone loss as observed by Micro-CT. Although the mechanism of the osteoclast inhibitory effect of Sirt6 was not confirmed in this study, it can be inferred from previous experimental studies. Sirt6 inhibited production of RANKL, a crucial cytokine for osteoclastogenesis, by regulating deacetylation of the RANKL promoter region in osteoblasts and osteocytes [22]. Additionally, Sirt6 promoted preosteoclast apoptosis by modulating the stability of estrogen receptor protein during osteoclastogenesis, thereby suppressing the number of osteoclasts [23]. The multi-mechanisms of Sirt6 in osteoblasts and osteoclasts that maintain bone homeostasis were thought to inhibit osteoclast numbers in ligature-induced periodontitis.

A Sirt6 activator with anti-inflammatory and anti-osteoclastogenesis properties acts by similar therapeutic mechanisms to host modulators such as monoclonal antibodies, against IL-6 receptor, specialized pro-resolving mediators (SPMs), and complement inhibitors that were recently reported to have periodontitis efficacy [29]. Host modulatory agents, which control the host inflammatory response to the microbial challenge, showed a positive impact on progression of periodontal disease by promoting resolution of inflammation and restoring tissue homeostasis, as well as mitigating the risk of periodontitis-related systemic comorbidities [29,30]. Considering the positive effects of host modulators for periodontitis in animal models and human clinical trials, the Sirt6 activator could be an adjunctive therapeutic strategy for periodontitis. Additionally, the SPMs resolvin E1 and lipoxin A4 have been reported to inhibit NF-κB activation via sirtuins (Sirt1, Sirt6, and Sirt7) in pulpitis, supporting the therapeutic potential of a Sirt6 activator in periodontitis [31].

To verify the role of Sirt6 in periodontitis, we established a ligature-induced periodontitis mice model using nylon sutures. At the ligation site, a plaque forms around the ligature with continuous inflammatory infiltration, leading to destruction of periodontal tissue within a few days [14,22]. Because the pathological outcome of periodontitis patients and the phenotype of this mice model are similar, this method has been widely used to understand the mechanisms of periodontitis to develop new drugs [28]. An animal model of ligation-induced periodontitis was applied to understand the mechanism of severe periodontitis that adversely affects organs such as the kidneys [32], ovaries [33], liver [34,35], arteries [36], and brain [37]. However, this study and several others did not observe systemic inflammatory responses or systemic osteoclast activity [36,38], possibly due to several factors. First, female mammals have more severe inflammatory responses against ligatures [39], but we used male mice. Second, although long-term ligation (10 weeks) induces a more severe systemic inflammatory response compared with short-term ligation (three weeks) [40], we performed experiments with short-term ligation. Third, we did not additionally apply other conditions (pathogen injection, high-fat diet, or genetic modulation) that could exacerbate systemic inflammation to the periodontitis model [36]. To confirm the role of Sirt6 in the periodontitis model that induces systemic inflammation or periodontitis-related systemic comorbidities, these factors should be included in the experimental conditions in future studies. This study offers the first elucidation of the role of Sirt6 in ligature-induced periodontitis, despite several limitations to explain the detailed mechanism of the protective effect of Sirt6 on alveolar loss.

## 4. Materials and Methods

### 4.1. Animal Experiments

Sirt6-overexpressing Tg mice (Sirt6Tg, C57BL/6-Tg (RP23-352G18)1Coppa/J) were obtained from the Jackson Laboratory (Bar Harbor, ME, USA). Periodontal ligation was performed on 12-week-old wild type C57BL/6J male mice and Sirt6Tg male mice. A 5-0 nylon suture was used as a ligature, and it was placed on the second left maxillary molar. The mice were sacrificed after one week or three weeks. For pharmacological Sirt6 activation, MDL-801 (Chemscene, Monmouth Junction, NJ, USA), allosteric Sirt6 activator was given to wild type C57BL/6J male mice via oral gavage (100 mg/kg) twice a week. MDL-801 compound was mixed with 5% DMSO, 10% Kohipher, 30% Trisol buffer (pH 7), and 55% 1X PBS. The vehicle group was treated with the same amount of the dissolving agent (5% DMSO, 10% Kohipher, 30% Trisol buffer (pH 7), 55% 1X PBS) as the MDL-801 group via oral gavage. To ensure Sirt6 activation of the MDL-801-treated group, MDL-801 was first administered approximately 2 weeks earlier than the ligature placement. MDL-801 was given regularly until the end of the third week of ligation. Mice had free access to food and water and were maintained in a room with controlled humidity (50%) and temperature (22 °C) on a 12-h light/dark cycle. All the protocols used in this study were approved by the Institutional Animal Care and Use Committee of Jeonbuk National University (permit no. JBNU 2022-0923).

### 4.2. Micro-CT Analysis

Micro-CT scanning was performed on left maxilla samples using a SKYSCAN 1076 Micro-CT (SkyScan, Kontich, Belgium) installed in the Center for University-Wide Research Facilities (CURF) at Jeonbuk National University. The X-ray source was set to a pixel size of 8.8 mm at 75 kV and 100 mA, and 400 projections were acquired over an angular range of 180 u. All specimens were applied at a constant threshold with a global thresholding algorithm. Voxels with intensities exceeding the threshold were considered to contain mineralized tissue. The threshold was the intensity corresponding to 45% of the average intensity of the maxilla bone in the specimens. The volume of interest (VOI) for each sample was adjusted using the software CTAn version 2.0. Using CT Analyzer, the top and bottom range of 4.48 mm were selected from raw images, and the same value was applied to every sample. The regions of interest (ROIs) were interpolated on sagittal images. ROIs specifically included alveolar bone regions surrounding each of the three maxillary molars. The density ranges of binary images were calibrated, and bone parameters were obtained from the three-dimensional analysis. Percent bone volume was calculated using the formula bone volume/tissue volume (BV/TV). Tb.Th is trabecular mean thickness, and Tb.N is the trabecular number accounting for the average number of trabeculae per unit length. Tb.Sp is the trabecular separation that is the mean distance between trabeculae. The overall degree of bone resorption was also presented in three-dimensional models using CT Vol. On CT Analyzer, the change in bone height of the second maxilla was measured from cement to enamel junction to alveolar bone crest (CEJ-ABC).

### 4.3. Histology

Maxillae were fixed in 10% formalin and decalcified in 10% EDTA for 10 weeks. After dehydration, they were embedded in paraffin and sectioned. The sectioned (5 μm) maxillae samples were then deparaffinized and stained with hematoxylin and eosin (H&E) and tartrate-resistant acid phosphatase (TRAP, Sigma-Aldrich, St. Louis, MO, USA) according to the manufacturer’s instructions. For immunohistochemistry, the sectioned maxillae samples underwent heat-induced antigen retrieval with citrate buffer (DAKO, Carpinteria, CA, USA). HRP/DAB detection IHC system (ab236466, Abcam, Cambridge, UK) was applied to samples according to the manufacturer’s instructions. As briefly described, after blocking, the samples were incubated overnight at 4 °C with primary antibodies as follows: Sirt6 (12486s, Cell Signaling Technology, Danvers, MA, USA, 1:200); IL-6 (600-1131, Novus Biologicals, Centennial, CO, USA, 1:250); IL-1β (12242, Cell Signaling Technology, Danvers, MA, USA, 1:250). After primary antibodies, the sections were incubated with HRP-goat anti-rabbit IgG secondary antibody. Then, DAB chromogen was applied, and sections were counter-stained with hematoxylin. Images were taken via a Leica DM750 microscope (Leica, Wetzlar, Germany). For immunofluorescence staining, the sectioned samples were incubated with blocking buffer (5% BSA) and then antibodies were applied as follows, F4/80 (MCA497GA, Bio-Rad, Hercules, CA, USA, 1:500); CD11b (101205, Biolegend, San Diego, CA, USA, 1:250). Staining was visualized using relevant secondary antibody conjugated to goat anti-rat Alexa Fluor 488 (Invitrogen, Carlsbad, CA, USA, 1:500) and DAPI (1 ug/mL; Burlington, MA, USA, Sigma) added to each sample. Fluorescence images were acquired with an LSM 980 Confocal Microscope (Carl Zeiss, Oberkochen, BW, Germany). Image intensity was analyzed with the software program Image J (Version 1.53t, NIH, Bethesda, MD, USA; http://imagej.nih.gov/ij/).

### 4.4. Western Blot

Resected maxillae were homogenized, and protein lysates were prepared. They were separated on 10% or 12% SDS-PAGE and transferred to PVDF membranes. After blocking with 5% non-fat dry skim milk, the membrane was incubated with primary antibodies overnight at 4 °C. Antibodies were used as follows: Sirt1 (ab110304), Sirt5 (ab195436, Abcam); Sirt2 (sc-28298), TNF-α (sc-52746), β-actin (sc-47778, Santa Cruz Biotechnology, Dallas, TX, USA); Sirt4 (3224) and Sirt7 (3099, Biovision, Waltham, MA, USA); IL-β1(12242), Sirt3 (5490s), Sirt6 (12486s, Cell Signaling Technology, Danvers, MA, USA,), IL-6 (600-1131, Novus Biologicals, Centennial, CO, USA). Then, secondary antibodies were added. Gel images were taken with a LAS-3000 Luminoimage analyzer (Fujifilm, Tokyo, Japan) and analyzed using the software program ImageJ (Version 1.53t, NIH, Bethesda, MD, USA; http://imagej.nih.gov/ij/).

### 4.5. Statistical Analysis

Data were expressed as mean ± standard error and were statistically analyzed using the software program GraphPad Prism 9.0.0. Comparisons between two groups were evaluated via a nonparametric Mann–Whitney U test or unpaired two-tailed Student’s *t*-test, and one-way ANOVA was applied to compare more than two groups, followed by Tukey’s test for multiple comparisons. A *p*-value < 0.05 was considered significant.

## 5. Conclusions

The present study identified the protective effect of Sirt6 on inflammatory alveolar bone loss. To verify Sirt6 functions in periodontal disease, a ligature-induced periodontitis mouse model was established and applied to Sirt6-overexpressing transgenic and drug-treated mice. Ligation on maxillary molars resulted in severe alveolar bone loss and inflammatory cytokine recruitment around the placement of ligation. These symptoms were significantly attenuated in Sirt6-overexpressing mice and Sirt6 activator-administrated mice. Taken together, genetic or pharmacologic activation of Sirt6 ameliorates ligature-induced periodontitis by suppressing inflammation and reducing the number and activity of osteoclasts. Therefore, our results suggest that a Sirt6 activator could be used as a potential drug for periodontitis treatment.

## Figures and Tables

**Figure 1 ijms-24-10714-f001:**
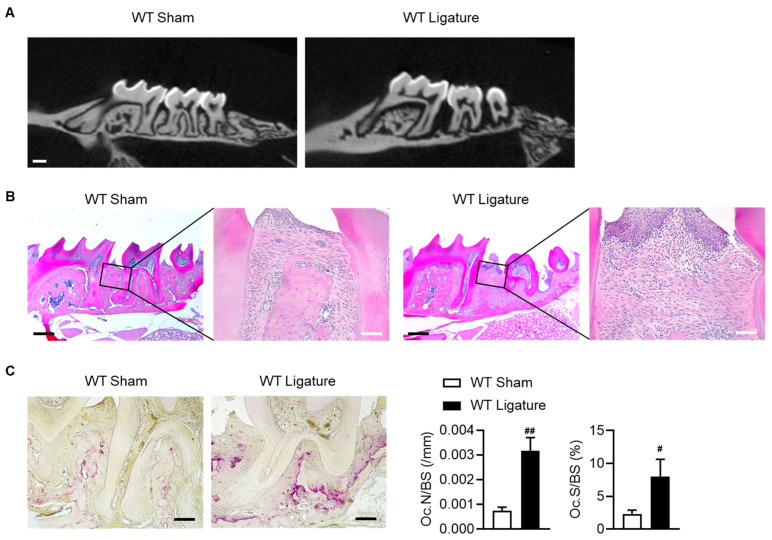
Establishment of inflammatory bone loss in ligature-induced periodontitis. Ligation using a 5-0 nylon suture on the second maxillary molar or sham surgery was applied to twelve-week-old WT mice for one or three weeks. We then performed micro-CT and histology. (**A**) Representative micro-CT image of the three-week ligatured maxilla. White bar = 500 μm. (**B**) Representative H&E staining of the three-week ligatured maxilla. Black bar = 500 μm and white bar = 50 μm. (**C**) Representative TRAP staining of the one-week ligatured maxilla. Black bar = 100 μm. Osteoclast numbers per bone surface (Oc.N/BS) and the ratio of osteoclast surface to bone surface (Oc.S/BS) were measured (*n* = 4–7), and the values are means ± SEMs. # *p* < 0.05, ## *p* < 0.01 versus WT sham; 2-tailed, nonparametric Mann–Whitney U test.

**Figure 2 ijms-24-10714-f002:**
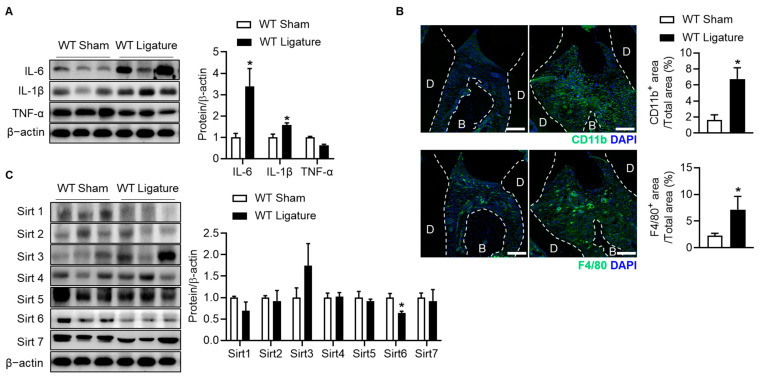
Reduction of Sirt6 expression in ligature-induced periodontitis. Protein lysate from three-week ligatured maxilla or sham surgery maxilla were analyzed using Western blotting. (**A**,**C**) Protein level of inflammatory cytokines and sirtuin proteins were determined. The band intensities were determined by densitometry (*n* = 3). (**B**) Representative immunofluorescence staining of CD11b positive neutrophils (green), F4/80 positive macrophages (green), and DAPI (blue). B; alveolar bone, D; dentin. The white dotted line indicates the boundary between alveolar bone and dentin and the gingiva. White bar = 100 μm. Signal intensities of CD11b and F4/80 in total area (gingiva tissue and periodontal ligament area) were calculated using an image analyzer (*n* = 4–5). And the values are means ± SEM. * *p* < 0.05 versus WT sham; unpaired two-tailed Student’s *t* test.

**Figure 3 ijms-24-10714-f003:**
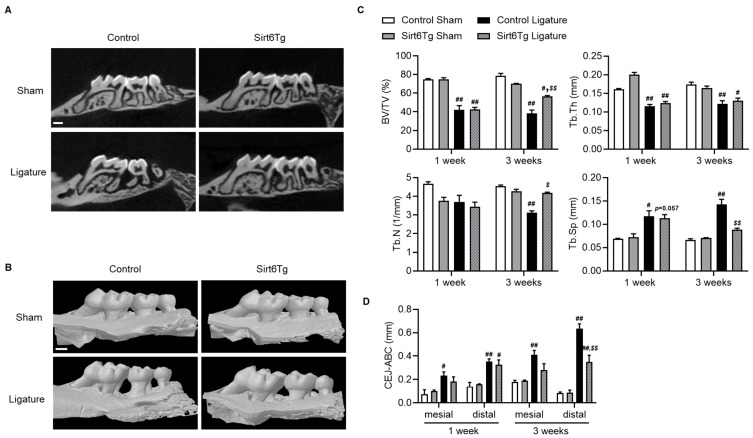
Attenuation of ligature-induced bone loss in Sirt6Tg mice. (**A**,**B**) Representative micro-CT image and 3D reconstruction image of the three-week ligatured maxilla or sham surgery maxilla in the control and Sirt6Tg mice. White bar = 500 μm. (**C**) Bone morphometric parameters were analyzed in the one-week ligatured and three-week ligatured maxilla (*n* = 3–5). BV/TV: bone volume/tissue volume; Tb.Th: trabecular thickness; Tb.N: Trabecular number; Tb.Sp: trabecular separation. (**D**) The length of the cemento–enamel junction (CEJ) to the alveolar bone crest (ABC) was analyzed in each group’s ligatured maxillary second molar (*n* = 3–5), and the values are means ± SEM. # *p* < 0.05, ## *p* < 0.01 versus sham; $ *p* < 0.05, $$ *p* < 0.01 versus control ligature; 1-way ANOVA with Tukey’s post hoc comparison.

**Figure 4 ijms-24-10714-f004:**
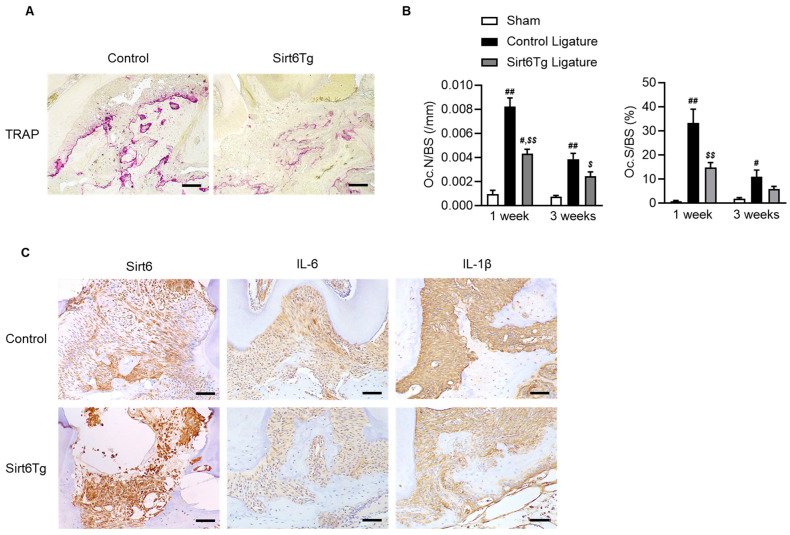
Reduction of osteoclast numbers and inflammation in Sirt6Tg mice. (**A**) Representative TRAP staining of one-week ligatured control and Sirt6Tg mice. Black bar = 50 μm. (**B**) Osteoclast numbers per bone surface (Oc.N/BS) and the ratio of osteoclast surface to bone surface (Oc.S/BS) were measured (sham: *n* = 3–6; ligature; *n* = 11–15), and the values are means ± SEM. # *p* < 0.05, ## *p* < 0.01 versus sham; $ *p* < 0.05, $$ *p* < 0.01 versus control ligature; 1-way ANOVA with Tukey’s post hoc comparison. (**C**) Representative immunohistochemical staining of Sirt6, IL-6, and IL-1β in controls and Sirt6Tg. Black bar = 50 μm.

**Figure 5 ijms-24-10714-f005:**
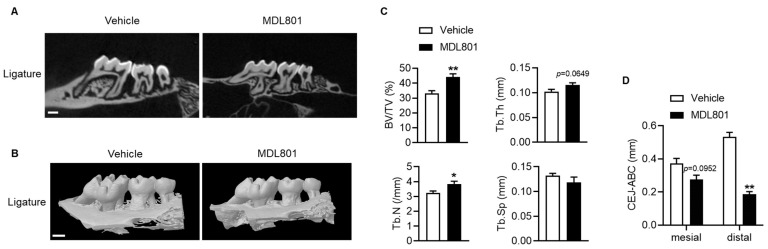
Amelioration of ligature-induced bone loss in Sirt6 activator-treated mice. (**A**,**B**) Representative micro-CT images and 3D reconstruction images of the maxilla of vehicle or MDL801 treatment mice three weeks after ligation. White bar = 500 μm (**C**) Bone morphometric parameters were analyzed for alveolar bone from vehicle or MDL801 treatment mice three weeks after ligation (*n* = 6). BV/TV: bone volume/tissue volume; Tb.Th: trabecular thickness; Tb.N: trabecular number; Tb.Sp: trabecular separation. (**D**) The length of the cemento–enamel junction (CEJ) to the alveolar bone crest (ABC) was analyzed in each group’s ligatured maxillary second molar (*n* = 5), and the values are means ± SEM. and the values are means ± SEM. * *p* < 0.05, ** *p* < 0.01 versus vehicle; 2-tailed, nonparametric Mann–Whitney U test.

## Data Availability

The data presented in this study are available on request from the corresponding author.

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
