# Peer review of "Sirt6 Activation Ameliorates Inflammatory Bone Loss in Ligature-Induced Periodontitis in Mice"

_ijms, 2023, doi:10.3390/ijms241310714_

Round 1

Reviewer 1 Report

This study describes a beneficial effect of Sirt6 on ligature-induced degradation of periodontal ligament and alveolar bone. The study is well-performed and well-written. The various analyses are clearly described and the data collected are convincing.

Comment:

1.       The authors analyzed expression of a number of inflammation-related cytokines in tissue obtained from control and “ligature” mice. Such an analysis gives some information on the presence or possible absence of an inflammatory response but does not tell the whole story. In addition the authors are urged to analyze the cellular response. Is there any effect on the cellular composition in the tooth-related tissues of these mice. This analysis is likely to provide crucial information on possible effects of Sirt6 on the cellular response.

2.       What was the control for the mice receiving MDL-801? Did control mice receive the dissolvent used to dissolve MDL-801 (MDSO, Kohipher and Trisol)?  

Reviewer 2 Report

Many thanks for the paper submission. The article deals with Sirt6 activation ameliorates inflammatory bone loss in ligature-induced periodontitis in mice. Laboratory research is always interesting in dentistry: however today most of the research is in vivo. This is a nice article, however some modifications are required.

1)at line 52 please modify the sentence into

"..Emerging evidence has suggested that severe periodontitis is associated with other chronic inflammatory diseases including diabetes, cardiovascular disease, atherosclerosis and surgical site infection..."

2) This manuscript highlights a possible role from Sirt6 in periodontitis, which adds to the subject area compared with other published material. Please follow this order: introduction; materials and methods; results; discussion and add 1 keyword and place them in alphabetical order.

4. Conclusion should be rephrased.

5. Some references regarding the role of bone loss prevention and concepts in introduction should be reported:

i) Please add the present reference:
Chisci G, Gabriele G, Gennaro P. Periodontal disease before and after fractures of the mandible. Br J Oral Maxillofac Surg. 2023 Jan;61(1):116. doi: 10.1016/j.bjoms.2022.09.016. Epub 2022 Nov 22. PMID: 36517341. 

(ii) at line 59 please modify the sentence into "Cumulated alveolar bone loss weakens the support structure of the teeth, causing tooth mobility and loss: many procedures are described in literature to reduce bone loss and alveolar shrinking after tooth extraction, however the prevention of tooth loss is the primary objective.
 Please cite the following:   Chisci G, Fredianelli L. Therapeutic Efficacy of Bromelain in Alveolar Ridge Preservation. Antibiotics (Basel). 2022 Nov 3;11(11):1542. doi: 10.3390/antibiotics11111542. PMID: 36358197; PMCID: PMC9687015.   Chisci G, Hatia A, Chisci E, Chisci D, Gennaro P, Gabriele G. Socket Preservation after Tooth Extraction: Particulate Autologous Bone vs. Deproteinized Bovine Bone. Bioengineering (Basel). 2023 Mar 27;10(4):421. doi: 10.3390/bioengineering10040421. PMID: 37106608; PMCID: PMC10136074.   Gürsoy UK, Gürsoy M, Könönen E. Biomarkers and Periodontal Regenerative Approaches. Dent Clin North Am. 2022 Jan;66(1):157-167. doi: 10.1016/j.cden.2021.06.006. Epub 2021 Jul 31. PMID: 34794552. 

6. Figures are ok

Round 2

Reviewer 2 Report

accept